# Nanoparticle-Based Plasmonic Biosensor for the Unamplified Genomic Detection of Carbapenem-Resistant Bacteria

**DOI:** 10.3390/diagnostics13040656

**Published:** 2023-02-09

**Authors:** Oznur Caliskan-Aydogan, Saad Asadullah Sharief, Evangelyn C. Alocilja

**Affiliations:** 1Department of Biosystems and Agricultural Engineering, Michigan State University, East Lansing, MI 48824, USA; 2Global Alliance for Rapid Diagnostics, Michigan State University, East Lansing, MI 48824, USA

**Keywords:** carbapenemase-producing bacteria, KPC, biosensor, GNP, colorimetric

## Abstract

Antimicrobial resistance (AMR) is a global public health issue, and the rise of carbapenem-resistant bacteria needs attention. While progress is being made in the rapid detection of resistant bacteria, affordability and simplicity of detection still need to be addressed. This paper presents a nanoparticle-based plasmonic biosensor for detecting the carbapenemase-producing bacteria, particularly the beta-lactam *Klebsiella pneumoniae* carbapenemase (*bla_KPC_*) gene. The biosensor used dextrin-coated gold nanoparticles (GNPs) and an oligonucleotide probe specific to *bla_KPC_* to detect the target DNA in the sample within 30 min. The GNP-based plasmonic biosensor was tested in 47 bacterial isolates: 14 KPC-producing target bacteria and 33 non-target bacteria. The stability of GNPs, confirmed by the maintenance of their red appearance, indicated the presence of target DNA due to probe-binding and GNP protection. The absence of target DNA was indicated by the agglomeration of GNPs, corresponding to a color change from red to blue or purple. The plasmonic detection was quantified with absorbance spectra measurements. The biosensor successfully detected and differentiated the target from non-target samples with a detection limit of 2.5 ng/μL, equivalent to ~10^3^ CFU/mL. The diagnostic sensitivity and specificity were found to be 79% and 97%, respectively. The GNP plasmonic biosensor is simple, rapid, and cost-effective in detecting *bla_KPC_*-positive bacteria.

## 1. Introduction

Infectious disease outbreaks have killed thousands of people with severe negative global economic impacts. Among these, antimicrobial-resistant (AMR) bacteria are a major growing concern [1]. Due to AMR infections, it is estimated that approximately 700,000 people die each year worldwide [2], with 39% of the cases linked to last-resort AMR bacteria [3]. According to a recent estimate, by the year 2050, AMR infections will result in 10 million deaths per year, along with losses worth USD 100 trillion [4].

Carbapenems, a subclass of β-lactam antibiotics, are the last line of defense against severe multi-resistant infections [1,5]. The incidence and spread of carbapenem-resistant bacteria have increased globally at an alarming rate since early 2010 [5]. Subsequently, carbapenem-resistant *Acinentobacter baumannii*, *Pseudomonas aeruginosa*, and carbapenem-resistant *Enterobacterales* (CRE) have been listed as critical priority pathogens by the World Health Organization (WHO) since 2017 [6] and the Centers for Disease Control and Prevention (CDC) since 2019 [7]. Particularly, CRE results in 1100 deaths and 13,100 infections in the USA [7], with a significant fraction of these infections potentially resulting in death due to a lack of alternative antibiotic treatments [5,7].

Carbapenem resistance mainly results from carbapenemase production and the porin gene mutation (non-carbapenemase). The presence of carbapenemase is usually sufficient for carbapenem resistance [8]. Production of carbapenemase can occur by mutations or through horizontal gene transfer (HGT) [9,10]. For example, a subset of bacterial cells from susceptible populations can develop spontaneous mutations in genes, allowing them to survive in the presence of antibiotics (carbapenem). These mutations lead to a selection of the genes that alter antibiotic actions through enzyme (carbapenemase) production [10,11], which are commonly transferable. In HGT, these carbapenemase genes are often located on mobile genetic elements, such as integrons, transposons, and plasmids, which contribute to their spread [1,12,13]. This results in carbapenemase-producing (CP) bacteria cases in humans who are not using the antibiotic or are hospitalized but interact with environments and hosts colonized with CP bacteria [2]. The most common enzymes are *Klebsiella pneumoniae* carbapenemase (KPC), Metallo-β-lactamases (New Delhi Metallo-lactamase (NDM), Imipenemase (IMP), and Verona integron-encoded Metallo-lactamase (VIM)), and Oxacillinase-48 (OXA-48). The CDC routinely tests these common carbapenemase types through the Antibiotic Resistance Laboratory Network [14].

In diagnostic protocols, antibiotic susceptibility tests (AST) are widely used in clinical and public health labs to assess antibiotic resistance profiles of target isolates [7]. AST methods usually require overnight culturing in selective media for species identification, followed by further growth in antibiotic solutions for determining their antibiotic-resistant profile [15,16,17,18]. Culture-based tests for identification of the CP bacteria include Modified Hodge Test (MHT) [19] and carbapenemase inactivation methods (CIM) [20,21], which are cost-effective and widely applicable. There have also been specific mediums designed for CP strain screening [22,23]. However, these culture-based tests are labor-intensive and require time-consuming steps for the isolation of pure cultures, taking days to weeks for determining their resistant profile [15,16,18].

Several phenotypic techniques have been developed to shorten the required time to detect CP bacteria based on their carbapenemase hydrolysis activity. For example, an ultraviolet (UV) spectrophotometric method has been developed to measure the imipenem hydrolysis activity of CP bacteria [24]. Further, the combination of automation and a user-friendly interface recently made matrix-assisted laser desorption/ionization time-of-flight mass spectrometry (MALDI-TOF MS) popular in clinical labs. This technique identifies bacteria at the species and genus level from single isolated colonies [25]. Bioluminescence-based assays have also been developed for the detection of CP bacteria within 2.5 h [26]. Recently, a colorimetric assay, the Carba NP test and its commercial kits, have been developed for simple and cost-effective detection within 2 h [20,21,25,27]. However, these assays require pure cultures and depend on the growth rate of bacteria [25]. Many of these rapid phenotypic techniques also require expensive and complex equipment, data analysis, and trained personnel, which reduces their applicability in low-resource settings [16,17,18,28].

Molecular methods have also been developed as effective techniques to detect specific resistant genes. PCR-based methods, DNA microarray and chips, whole genome sequencing (WGS), loop-mediated isothermal amplification (LAMP), and fluorescence in situ hybridization (FISH) are used as genotypic techniques for the detection of resistance [17,18,28]. Among these, PCR-based approaches have been used as a gold standard for detecting β-lactam (*bla*) resistant genes. For example, multiplex PCR has been developed to detect 11 acquired carbapenemase genes using three different multiplex reaction mixtures [29]. Numerous automated systems have also been designed to detect and confirm the presence of target genes for bacteria and their resistance [21,25], although associated costs limit their use [25]. LAMP has been used as an alternative to PCR, particularly in low-resource settings, due to its simplicity and cost-efficiency. However, it still has limitations such as complex primer design [17,18,30]. Many genotypic methods offer rapid detection with higher sensitivity and specificity, but require costly reagents, equipment, and need skilled operators [16,18].

Biosensors have emerged as an alternative approach for simple, rapid, and low-cost detection. These analytical devices utilize biological or chemical reactions and convert the recognition event into measurable signals to detect target analyte [31,32]. A few biosensor studies have been developed for CP bacteria detection. For instance, a label-free electrochemical biosensor detected PCR amplified *bla_NDM_* gene using impedance spectroscopy [33]. Other voltammetry-based techniques have allowed for *bla_KPC_* detection in 45 min [34]. Recently, the Surface-Enhanced Raman Scattering technique (SERS) for carbapenem resistance detection has also been developed using silver nanoparticles [35], gold nanostars [36], and gold and silver nanorods [37] with higher specificity and sensitivity. However, these techniques require multivariate data analysis for detection. In another study, a lateral flow biosensor was developed to identify and differentiate *bla_OXA-23-like_*, by multiple cross displacement amplification [38]. Elsewhere, plasmonic nanosensors were developed using GNPs for colorimetric detection of CP pathogens based on carbapenemase activity and pH changes [39]. However, these biosensors are still costly and require complex techniques for signal measurements and data analysis [28]. Colorimetric gold nanoparticle (GNP) biosensors can offer an ideal solution.

The size-dependent chemical, electric, optical, and physical properties of GNPs are their primary advantages [40,41,42]. In addition, GNPs are chemically stable and can be easily modified for biosensor applications, stimulating their popularity and use in recent years [41,42,43]. They have unique optical properties; free electrons of GNPs in colloidal solutions induce coherent oscillation once the frequency of light matches with electron frequency, producing a strong SPR band. The SPR band is strongly distance-dependent and transforms depending on monodispersed or aggregated state of GNPs in a liquid solution, resulting in a visible color change [40,41,42]. Small monodispersed GNPs have an SPR absorption peak at around 520 nm, with a red color, which shifts to longer wavelength with aggregation of particles leading to color change to blue or purple [43,44,45]. Compared to other nanoparticles displaying SPR, which are unstable and toxic, GNPs are stable for a long time, making them cost-effective [46]. The colorimetric nature of this platform is noteworthy, offering rapid and simple visual detection in under one hour without complex and costly equipment [44,47,48].

GNPs have been extensively used to detect target DNA from several bacteria. For instance, thiol-capped GNPs were used to detect *Klebsiella pneumoniae* within one hour using amplified *K2A* gene [47] and unamplified DNA of uropathogenic *E. coli* [49]. However, traditional GNP synthesis and probe functionalization techniques can require labor and take several days [48,49]. Rapid synthesis and functionalization of GNPs are preferable, and dextrin-coated GNPs have shown higher stability and biocompatibility [50]. These GNPs have been used earlier to detect unamplified DNA of *E. coli* O157:H7 [44] and *Pseudoperonospora cubensis* [50] within 30 min. However, their application for detecting AMR including CP bacteria has not been documented.

Rapid detection is a significant step to control and prevent the emergence and spread of infections. For instance, the mortality rate of CRE infections rises by approximately 8% for each hour of delay in obtaining a correct diagnosis and suitable antibiotic treatment [51]. The delayed appropriate therapy in CP-CRE increase the risk of mortality from 0.9% to 3.7%, hospital cost from ~USD 10,000 to ~USD 25,000, and hospital stay from 5.1 days to 8.5 days [34,52]. Thus, this study proposes a rapid and simple plasmonic biosensor platform for detecting CP-bacteria, specifically the KPC-producing bacteria. The KPC is the most prevalent enzyme type among CP bacteria in the US and the world [10,53].

### Novelty of This Study

The GNP-based plasmonic biosensor was designed to detect KPC-producing bacteria. This assay only requires dextrin-coated GNPs, DNA probes (*bla_KPC_*) for target bacteria, and DNA samples. Colorimetric results were quantified using absorbance spectra measurements by UV-Vis spectrophotometer. This study’s novelty is in the following aspects: (1) our GNP plasmonic biosensor using DNA probe is the first study in detecting AMR, including CP bacteria, (2) probe-functionalization protocols are not required, and (3) PCR amplification is not required for detection. Figure 1 briefly describes the generalized procedure of the plasmonic biosensor employed in this study.

## 2. Materials and Methods

### 2.1. Materials

A total of 47 bacterial cultures were used in this study: 3 bacteria from the American Type Culture Collection (ATCC), 38 carbapenemase-producing (CP) bacteria isolates from the Michigan Department of Health and Human Services (MDHHS), and 6 bacteria from Dr. Evangelyn Alocilja’s Nano-Biosensors Laboratory at Michigan State University (MSU). DNA extraction kits were purchased from Qiagen (Germantown, MD, USA). NanoDrop One from ThermoFisher Scientific (Waltham, MA, USA) was used to quantify DNA samples and absorption spectra. Oligonucleotide probes were ordered from Integrated DNA Technologies (IDT; Coralville, Iowa). Tryptic Soy Agar (TSA) and Tryptic Soy Broth (TSB), Hydrochloric acid (HCl), gold (III) chloride (HAuCl_4_), sodium carbonate (Na_2_Co_3_), 11-mercaptoundecanoic acid (MUDA HS(CH_2_)_10_CO_2_H), sodium dodecyl sulfate (SDS, C_12_H_25_NaO_4_S), dextrin from potato starch were purchased from Sigma Aldrich (St. Louis, MO, USA).

### 2.2. Bacterial Cultures

Bacterial strains of *E. coli* C-3000 (15597), KPC-producing carbapenem-resistant *E. coli* (BAA-2340), and *Klebsiella pneumoniae* subsp. *pneumoniae* (13883) were obtained from ATCC. Frozen cultures of *Salmonella enterica* serovar Typhimurium, *Salmonella enterica* serovar Enteritidis, *Klebsiella pneumoniae*, and *Enterobacter cloacae*, and carbapenem-resistant *Klebsiella pneumoniae* were obtained from MSU. The CP bacteria isolates from MDHHS included 12 KPC-producing bacteria: *E. coli* (2), *E. cloacae* (1), *K. pneumoniae* (3), *K. aerogenes* (2), *Raultella ornithinolytica* (2)*, Citrobacter amalonaticus* (1), *Citrobacter freundii* (1), and 26 non-KPC (IMP, NDM, OXA-48, VIM)-producing bacteria: *E. coli* (5), *K. pneumoniae* (4), *E. cloacae* (7), *K. oxytoca* (1), *C. freundii* (2), *Providencia rettgeri* (2), *Proteus mirabilis* (2), *Morganella morganii* (2), and *P. aeruginosa* (1). Carbapenem-resistant bacteria isolates from MDHHS were verified by molecular (CARBA-R Cepheid assay and CDC laboratory-developed assay) and growth-based AST methods.

Stock cultures of all isolates were stored at −80 °C. The cultures were refreshed by plating on TSA and incubated at 37 °C for 24–48 h. A single colony of the fresh bacterial cultures on TSA was then transferred into 9 mL of TSB with an overnight incubation at 37 °C before the experiment. The susceptible profile of *E. coli* C-3000, *S*. Typhimurium, *S*. Enteritidis, *E. cloacae*, *K. pneumoniae*, and *K. aerogenes* were confirmed using agar-dilution test (AST) [54].

### 2.3. DNA Extraction

The DNA of the pure bacteria cultures after overnight incubation was extracted using the commercial kit, which removes any interfering materials and was finally suspended in elution buffer (pH 8). The DNA concentration and quality were measured with the NanoDrop. DNA samples with acceptable A_260_/A_280_ and A_260_/A_230_ ratios, between 1.8 and 2.2, were used for the designed biosensor assay.

### 2.4. Probe Design and PCR Confirmation

A single-stranded oligonucleotide primer and probe were designed to target specifically KPC-producing bacteria. The primers and probe were designed using the *bla_KPC_* gene sequence of carbapenem-resistant *E. coli* (ATCC-2340), utilizing the design tools from NCBI BLAST (National Center for Biotechnology Information Basic Location Alignment Search Tool). Here, E-values were checked to indicate that the gene sequence is specific. The following aminated probe was used: 5′ CGG TGT GTA CGC GAT GGA TAC CGG CTC AGG CGC AAC TGT AAG TTA CCG CGC TGA GGA GCG. The following PCR primer sequence was used: F- 5′ CGGTGTGTACGCGATGGATA and R- 5′ TCCGGTTTTGTCTCCGACTG. The absence and presence of the *bla_KPC_* gene in all samples were confirmed by PCR; the protocol was adapted from an earlier study [29]. Amplified products were run on a 2% agarose gel in Tris Acetate EDTA (TAE) buffer at an applied voltage of 120 V for 1 h.

### 2.5. GNP Synthesis and Surface Modification

Dextrin-coated gold nanoparticles (GNPs) were synthesized using the alkaline synthesis method according to the procedure developed previously [55]. Briefly, gold (III) chloride trihydrate was dissolved in water and neutralized with sodium carbonate. Then, dextrin was added and heated at 150 °C under continuous stirring conditions until the solution turned wine red. The synthesis of GNPs was then confirmed by determining their absorption maxima using the NanoDrop at around 520 nm (red color). The GNPs were modified with 25 μM mercaptoundecanoic acid (MUDA) and suspended in 0.1 M borate buffer. As the MUDA-coated GNPs have -COOH groups, they create non-covalent interactions with amine groups on the aminated probe, leading to almost instantaneous GNP-probe functionalization. Batches of the surface-modified, ready-to-use GNPs were stored at 4 °C until further use. Since the GNPs are stable for a long time, new synthesis is not required for everyday analysis.

### 2.6. Biosensor Design and Optimization

The GNP-based plasmonic biosensor assay was developed with the following procedure [44]. Each biosensor trial included the extracted DNA sample (10 µL), 25 μM DNA probe (5 µL), and surface-modified GNPs (5 µL) in a single tube. Samples were then placed in a thermocycler to allow denaturation at 95 °C for 5 min, annealing at 55 °C for 10 min, and cooling to room temperature. This cycle enables target DNA to hybridize with the probe-GNP. Next, 0.1 M HCl was added to the sample, inducing GNP aggregation by distributing the electrostatic repulsion from the GNPs. However, target DNA bound to the GNP-probe prevents GNPs from aggregation. Thus, samples with target DNA remained red, while non-target samples allowed GNP aggregation, resulting in color change (purple or blue). The visual change in color of the GNPs was quantified by measuring their absorbance spectra in the wavelength range of 400–800 nm. Target samples were expected to have maximum absorbance at ~520 nm, while blue/purple samples shifted right, with higher absorption maxima. Quantification of the GNP aggregation was determined using absorbance ratios at 625 nm and 520 nm (A_625/520_), which is based on an earlier reported study [50].

The GNP biosensor optimization variables included the amount of HCl (5–10 µL) and the response time between HCl addition and reading the colorimetric results (5–10 min). The optimum HCl amount and aggregation time were determined through qualitative and quantitative analysis. Different amounts of 0.1 M HCl (5–10 μL) were separately added to the negative control (nuclease-free water), positive sample (10 ng/µL of KPC-producing *E. coli* BAA-2340), and negative sample (10 ng/µL of *E. coli* C-3000). Tubes were incubated until aggregation of negative samples without aggregation of the positive sample, which was visually observable. Absorbance spectra readings were taken at 5 min intervals after HCL addition. Readings were statistically analyzed at a 95% confidence interval; the optimized procedure had a significant and consistent difference between positive and negative samples, with a visible color change.

### 2.7. Limit of Detection Testing

The analytical sensitivity test was conducted at different DNA concentrations to determine the minimum detectable concentration of DNA. In this test, target and non-target DNA samples were serially diluted to lower concentrations, ranging from 20 to 1 ng/µL. Then, the target DNA sample (KPC-producing *E. coli* (BAA-2340)) was compared with a non-target sample (*E. coli* C-3000) at the same concentrations with a series of nine trials. Their visual color change and absorbance spectra measurements were used to determine the difference in GNP aggregations between the two samples. The A_625/520_ values were statistically analyzed at a 95% confidence interval.

### 2.8. Diagnostic Sensitivity and Specificity Testing

The biosensor was tested with a total of 47 DNA samples: 14 KPC-producing bacteria (target DNA) and 33 non-KPC-producing bacteria (non-target DNA), as listed in Table 1. A DNA concentration of 10 ng/µL was used for all samples with a series of nine trials. Each specificity trial included a negative control (DNA-free), target, and non-target samples. Their absorbance spectra measurements and images were collected during the experiment. Differences in A_625/520_ values among target and non-target samples were statistically analyzed at a 95% confidence interval.

The diagnostic sensitivity and specificity of this assay were calculated as described in an earlier study [56]. The sensitivity is the proportion of positive tests (True positive/(True positive + False negative)), and specificity is the proportion of negative tests (True negative/(True negative + False positive)).

### 2.9. Statistical Analysis

Data were presented with averages and standard deviations in bar graphs. The A_625/520_ values were compared among target and non-target samples using one-way analysis of variance (ANOVA) followed by Tukey’s HSD (honestly significant difference) test at a 95% confidence interval.

## 3. Results and Discussion

### 3.1. Characterization of GNPs and Principle of the GNP-Based Plasmonic Biosensor

Dextrin-coated GNPs used in the study were synthesized using an alkaline synthesis route. Successful GNP synthesis was confirmed by their wine-red appearance and a peak of maximum absorbance at ~520 nm, as seen in Figure 1a. Earlier reports have stated that the red color of GNPs and the absorbance peak at 520 nm indicate their size to be 10–50 nm in diameter [57]. For their biosensor application, the GNPs were surface-modified with MUDA, enabling their instantaneous non-covalent interaction with the aminated DNA probe [44]. The surface-modified GNPs were not affected by this modification, confirmed by their absorbance spectra (Figure 1a). Following sample DNA addition to GNPs and probe, and placement in the thermocycler, the absorbance spectra were again measured to confirm the stability of the GNPs. As seen in Figure 1a, GNPs did not show a shift in wavelength of absorption maxima, indicating that heating cycle or presence of DNA, did not affect the size of nanoparticles. After confirming GNPs’ stability, proof-of-concept of the assay was conducted.

As stated earlier, our plasmonic biosensor concept is based on the SPR of GNPs, which can be determined spectrophotometrically. GNP aggregation results from the distribution of the electrostatic repulsion, leading to a shift in their absorption maxima due to the distance-dependent nature of the SPR [44]. The shift in absorption maxima is associated with the color change from red to blue or violet. Thus, this study utilized GNPs’ absorbance spectra at 520 nm and shift in the peak of maximum absorbance following HCl addition for DNA detection. It was hypothesized that DNA-probe-GNPs’ conjugation would protect GNP against aggregation, leading to the maintenance of red color in the target DNA sample. In non-target samples, GNP would agglomerate due to a lack of protection by target DNA, changing their color from red to blue or purple. The biosensor thus produced a qualitative and quantitative signal, as seen in Figure 1b. The quantifiable signal corresponds to the presence or absence of the target analyte.

Optimization of the plasmonic biosensor using the *bla_KPC_* probe was initially conducted and resulted in 9 μL of 0.1 M HCl and a 5 min response time for further analysis. It should be noted that the GNPs are prevented from aggregation for at least 30 min. All steps from sample preparation to colorimetric analysis can be completed in approximately 30 min; this short duration is due to instantaneous GNP-probe functionalization.

### 3.2. Limit of Detection of the Plasmonic Biosensor

The lowest detection limit of the biosensor for KPC-producing bacteria was determined using the target DNA of KPC-producing *E. coli* (BA-2340) and the non-target DNA of susceptible *E. coli* C-3000. Both target and non-target DNA samples were diluted from 20 to 1 ng/μL by a factor of two. Target and non-target DNA samples at the same concentrations were compared using absorbance spectra measurements; tube images are shown in Figure 2. Quantification of GNP aggregation was achieved using A_625/520_. The difference in average A_625/520_ of target and non-target samples was higher at 20 ng/µL and 10 ng/µL followed by 5 ng/µL and 2.5 ng/µL. The least difference between the target and non-target samples was observed at 1 ng/μL. Statistically significant differences between target and non-target samples for each concentration were assessed using ANOVA followed by Tukey’s test. The target samples at 20 ng/µL, 10 ng/µL, 5 ng/µL, 2.5 ng/µL, and 1 ng/μL were significantly different from their non-target samples (*p* < 0.05). However, the average A_625/520_ of the target sample at 1 ng/μL overlapped with non-target samples at 20 ng/µL, 10 ng/µL, 5 ng/µL, 2.5 ng/µL, which were all similar (*p* > 0.05). Further, visual detection at 1 ng/µL was also not clearly observed. Thus, the detection limit was found to be 2.5 ng/µL.

Sensitivity results also indicate that the A_625/520_ values of non-target were in a similar range while those of target at various DNA concentrations followed a linear trend. Here, the A_625/520_ of target DNA samples at 20 ng/µL and 10 ng/µL were similar (*p* > 0.05), indicating that DNA concentrations above 10 ng/µL would have a similar absorbance ratio. Previous literature confirmed this observation where a GNP biosensor detected *E. coli* O157 with a similar absorbance peak in target samples at 10 ng/µL and 20 ng/µL [44]. Thus, our plasmonic biosensor can detect and differentiate target and non-target samples at and above 2.5 ng/µL.

The detection limit of our biosensor using unamplified DNA was lower than that of similar colorimetric assays in the literature. For instance, a colorimetric assay using thiolated GNPs for detecting *Staphylococcus epidermis* was achieved with a limit of 20 ng/µL [58]. In another study, *Leishmania* sps. were detected using multiple probes with DNA as low as 11.5 ng/µL [59]. Among other bacteria, uropathogenic *E. coli* was detected from the pure culture with a detection limit of 9.4 ng/μL [49], and *E. coli* O157 was detected with a limit of 2.5 ng/µL [44]. In other examples of GNP-based colorimetric assays, which had a significantly lower detection limit in clinical isolates, amplification of DNA samples was needed. Examples include the detection of *Acinetobacter baumannii* [60] and *Mycobacterium tuberculosis* [61] within 2 h following PCR. Compared to similar assays, the detection limit of our biosensor is reasonable without any PCR amplification and probe-functionalization and can be achieved in 30 min.

The detection limit of our biosensor corresponds to approximately 10^3^ CFU/mL. Other studies using lateral-flow immunochromatographic assay and colloidal GNPs successfully detected OXA-48 variants but demonstrated a higher detection limit at 10^6^ CFU/mL [62]. Similarly, electrochemical biosensor platforms for *bla_KPC_* detection [34] and plasmonic sensors for CP bacteria detection have all shown detection limits >10^4^ CFU/mL [39], with other methods requiring overnight culturing [19,23,25,35,63,64]. Our plasmonic biosensor promises a lower detection limit from unamplified DNA samples, along with a simple, rapid, and cost-effective detection.

### 3.3. Diagnostic Sensitivity and Specificity of the Plasmonic Biosensor

A total of 47 isolates were first confirmed for the presence of the *bla_KPC_* gene in the target (14) and its absence in non-target samples (33) by PCR amplification. Our plasmonic biosensor was then tested on all target and non-target samples. The PCR and biosensor results, with average of A_625/520_ and tube images are shown for all 47 samples in Figure 3, Figure 4, Figure 5, Figure 6 and Figure 7.

Figure 3a shows results from *E. coli* samples with three KPC-producing targets and six non-KPC-producing non-target samples. The plasmonic biosensor successfully detected two of three target samples. Sample spectra are also shown in Figure 3c. Among KPC-producing *E. cloacae* samples, the target was successfully detected, while none of the non-KPC-producing samples were detected by the biosensor (Figure 4a). No cross-reactivity with non-target DNA was observed, resulting in GNP agglomeration. For KPC-producing *K. pneumoniae*, all the target samples were successfully detected. However, one (NDM-producing) of six non-target samples was also detected by the biosensor (Figure 5a). The biosensor was also successful in detecting one of the two KPC-producing *K. aerogenes* samples and differentiated it from non-KPC producing carbapenem-resistant *K. aerogenes*, *K. oxytoca*, *P. rettgeri*, *P. aeruginosa*, and susceptible *K. aerogenes*, *S*. Enteritidis, and *S*. Typhimurium (Figure 6a). Lastly, our biosensor detected and differentiated KPC-producing *R. ornithinolytica*, *C. amalonaticus*, and *C. freundii* from non-KPC-producing carbapenem-resistant *C. freundii*, *M. morganii*, *P. mirabilis* (Figure 7a). The A_625/520_ values of all detected samples were significantly different than those of the non-detected samples (*p* < 0.05).

Overall, the designed DNA-based plasmonic biosensor used the *bla_KPC_* probe to successfully detect and differentiate KPC-producing bacteria from non-KPC-producing bacteria regardless of bacterium types. The rapid biosensor was in almost perfect agreement with the result of PCR amplification as seen in Figure 3b, Figure 4b, Figure 5b, Figure 6b and Figure 7b. The biosensor detected 11 of 14 target DNA samples and one of 33 non-target DNA samples, confirmed statistically. The average A_625/520_ of the detected target DNA samples in all trials was in the range of 0.43–0.63, while the absorbance ratio of non-target DNA detection was between 0.71–1.2 after HCl application (5 min response). The differences in absorbance ratio of all detected samples were significantly (*p* < 0.05) different than those of all non-detected samples.

The diagnostic specificity (true negative) and sensitivity (true positive) of our biosensor were found to be 97% and 79%, respectively (Table 2). These results were in the range of sensitivity and specificity levels of the phenotypic techniques used in clinical labs to detect carbapenem-resistant bacteria. For instance, MALDI-TOF MS detected the resistant bacteria with a range of 72.5–100% sensitivity and 98–100% specificity, along with an issue on OXA-48 identification [20,25]. The sensitivity of the Carba NP test was 73–100%, but it performed poorly in the detection of OXA-48 enzyme type [20,21,27]. Different types of commercial kits: RAPIDEC Carba NP test (first commercial test), β-CARBA test, Rapid CARB screen, Rapid Carb Blue kit, and Neo-CARB kit have been used to detect CP bacteria in the range of 15 min to 2 h with varying sensitivity (89.5–99%) and specificity (70.9–100%) from the pure culture [21,25]. In culture-based methods, sensitivity of the MHT was found to be 69% [65] and 93–98% [20], along with lower performance in the detection of NDM enzymes [20,21]. Lastly, the sensitivity of the CIM method was 98–100% [20,66], and 96.1% [66]. However, the CIM method is labor-intensive and takes days to determine the resistant profile of the bacteria [18,20].

Among the genotypic methods, the multiplex oligonucleotide ligation-PCR procedure helps to detect β-lactamase genes and their variations with high sensitivity and specificity (100% and 99.4%) in 5 h [25]. The LAMP method using hyfroxynaphtol blue dye (LAMP-HNB) and microarray techniques has also been used to detect carbapenemase enzymes with higher specificity and sensitivity at 100% and >90%, respectively [18,21]. The RNA-targeted molecular approach, NucliSENS EasyQKPC test, successfully detected *bla_KPC_* variants within 2 h, achieving 93.3% sensitivity and 99% specificity [67]. While genotypic methods show higher sensitivity and specificity, they are expensive and therefore limited in their field-application. Alternatively, our GNP biosensor offers cost-effective and rapid detection with good sensitivity and specificity for KPC-producing bacteria.

### 3.4. Improving GNP Plasmonic Biosensor Applicability and Accessibility

Rapid diagnostic techniques with higher specificity and sensitivity help improve diagnosis, disease management, epidemiology, and outbreak investigations [7]. As AMR infections, particularly carbapenem-resistant infections, are a global concern, rapid detection of the causative bacteria is of utmost importance. The GNP-based plasmonic biosensors can become an accessible and rapid detection method, especially in low-resource settings.

This plasmonic biosensor is the first study to detect AMR with KPC-producing bacteria. This study showed the potential of the plasmonic biosensor applicability in detecting the gene of KPC-producing bacteria in a short time. While the designed biosensor is easy, rapid, and cost-effective, its applicability and accessibility can be further developed through future works. For instance, visual results can also be quantified without requiring absorbance spectra measurements. Smartphone imaging techniques have been used to differentiate the color difference between target and non-target samples [39,68] and can be applied to this assay. Further, our biosensor using the *bla_KPC_* probe can be tested on more bacteria isolates to improve its sensitivity and specificity. The biosensor design can be extended to detect other carbapenemase genes using *bla_NDM_*, *bla_OXA-48_*, *bla_VIM_*, and *bla_IMP_* probes for broad-range detection. A multiplex GNP biosensor using multiple probes can be designed to detect all carbapenemase genes. The GNP biosensor can be further tested on clinical, environmental, and food samples to increase its real-world applicability. Even though carbapenems are used in human medicine, environmental, microbiological, and clinical investigations show that CP bacteria in the environment (water or soil ecosystems) could widely spread among animals and agricultural products [1,69,70,71,72,73]. Thus, rapid detection of CP bacteria, regardless of their pathogenicity, and its implementation in surveillance programs is important to prevent and control possible future endemics or pandemics.

Besides carbapenem resistance detection, our platform can be extended to other antibiotic-resistant bacteria (colistin, ampicillin, ESBL, etc.). Detection of specific genes with the plasmonic biosensor can be achieved in a simple, rapid, and cost-effective manner allowing it to be applied as an effective screening test in low-resource settings.

The colorimetric nature of this plasmonic biosensor offers rapid and simple visual detection in 30 min. The GNPs are easily prepared and modified and chemically stable for a long time. This assay has only three steps—adding extracted DNA to GNPs, placing the mixture in a thermocycler (acting as a heating block, not for DNA amplification), and HCl application. The estimated material cost of this GNP plasmonic biosensor is affordable at less than USD 2 per test, compared to rapid molecular methods (USD 23–150) and phenotypic methods (USD 2–10) [25]. This GNP biosensor does not require PCR amplification, complex and costly equipment such as a spectrophotometer, mass spectrophotometer or qPCR, and data analysis. Other inexpensive phenotypic methods require overnight culture and are therefore not rapid. Hence, our platform is affordable, rapid, and applicable; this designed plasmonic biosensor can be applied for point-of-care testing and field studies.

## 4. Conclusions

Dextrin-coated GNPs were used to design a DNA-based plasmonic biosensor to detect KPC-producing carbapenem-resistant bacteria in 30 min. This biosensor using the *bla_KPC_* probe can successfully visually differentiate between target (KPC-producing bacteria) and non-target samples (non-KPC-producing bacteria). Successful detection was achieved as low as 2.5 ng/µL (~10^3^ CFU/mL) from unamplified DNA samples. The diagnostic sensitivity and specificity were found to be 79% and 97%, respectively. In future works, this biosensor can be extended to detect different carbapenemase genes in clinical and biological samples.

## Data Availability

The data presented in this study are available upon request from the corresponding author.

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
