# Peer review of "Nanoparticle-Based Plasmonic Biosensor for the Unamplified Genomic Detection of Carbapenem-Resistant Bacteria"

_diagnostics, 2023, doi:10.3390/diagnostics13040656_

Round 1

Reviewer 1 Report

The present manuscript presents a nanoparticle-based plasmonic biosensor for detecting the carbapenemase-producing bacteria, particularly the beta-lactam Klebsiella pneumoniae carbapenemase (blaKPC) gene. The present sensor was synthesized using dextrin-coated gold nanoparticles (GNPs) and an oligonucleotide probe specific to blaKPC to detect the target DNA in the sample within a short time (30 min). The GNP-based plasmonic biosensor was tested in 47 bacterial isolatesand the the authors claims the stability of the coated-GNps. The manuscript generally is well written and the results are discussed in details and it consider as a good contribution in the field. However, a minor revision is needed to improve the manuscript.

1) The introduction is very long and could be shortened.

2) How long that the biosenor is stable and could be used or it is necessary to prepare new gold nanoparticles for eveyday analysis.

3) As the surface plasmon resonance is dependent on the metal type, size, shape, dielectric constant of the gold nanoparticles and the medium, what is the effect of interference in the medium.

4) In Figure 3, Plasmonic biosensor and PCR results of E. coli samples: (a) the average absorbance ratio at 625 and 520 nm (A625/520) of the target (T) and non-target (NT) samples with colorimetric results in tubes. Could the authors provide to this figuer the absorbance spectra of target (T) and non-target (NT) samples that the authors used for calculation of the ration at 625/520 nm.

Author Response

Thank you for your comments, the authors appreciate the feedback and input. All comments were accepted and appropriate changes with line numbers are listed below. The manuscript was run through Grammarly for spell-checking.

Question 1) The introduction is very long and could be shortened.

Response: The introduction has been shortened and word count was reduced from 1705 to 1494. However, to address comments from other reviewers, some information was added.

Question 2) How long that the biosenor is stable and could be used or it is necessary to prepare new gold nanoparticles for eveyday analysis.

Response: The gold nanoparticles are stable for at least 3 years under refrigerated conditions. Hence,  new synthesis of GNPs is not required for everyday analysis. This is now added in Lines 254-256.

Question 3) As the surface plasmon resonance is dependent on the metal type, size, shape, dielectric constant of the gold nanoparticles and the medium, what is the effect of interference in the medium.

Response: In our biosensor, DNA was extracted with kits which remove any interfering materials and suspended in elution buffer (pH 8). This is added in lines 227-228.

Question 4) In Figure 3, Plasmonic biosensor and PCR results of E. coli samples: (a) the average absorbance ratio at 625 and 520 nm (A625/520) of the target (T) and non-target (NT) samples with colorimetric results in tubes. Could the authors provide to this figuer the absorbance spectra of target (T) and non-target (NT) samples that the authors used for calculation of the ration at 625/520 nm.

Response: The absorbance spectra of Figure 3a is now added as Figure 3c, line 419-420.

Reviewer 2 Report

The manuscript is well-documented and should be of great interest to the readers who study on antimicrobial resistance. 

Author Response

Thank you for your comments, the authors appreciate the feedback. The manuscript was run through Grammarly for spell-checking.

Reviewer 3 Report

Main comments

   This manuscript describes about the development of nanoparticle based biosensor for the detection of carbapenemase resistant bacteria. The sensity and specificity were found to be 79% and 97%, respectively. This simple and rapid diagnostic system appear to have potential and limitation in Antibiotoic resistant diagnostic market and further encourage it to develop diagnostic systems other than carbapenem. Author need to address few questions before the acceptance of the manuscript.

Questions

1.     Please add little mechanism details behind how mutation in Carbapenemase gene cause the resistance to Carbapenem antibiotics in page 2, line 46?

2.     Author used Gold nanoparticle in this work. How is it cost effective compared to other metal nanoparticle?

3.     Can author mentioned how long the GNPs aggregation time is prevent in presence of target gene and maintain it’s colour?

4.     Based on the method described in the manuscript, this diagnostic tool require many pre-preparation steps such whole genome extraction, chemicals preparation to stabilize GNP. How it will be cost effective than other existing methods?

Author Response

Thank you for your comments, the authors appreciate the feedback and inputs. All comments were accepted and appropriate changes with line numbers are listed below. The manuscript was run through Grammarly for spell-checking.

Question 1) Please add little mechanism details behind how mutation in Carbapenemase gene cause the resistance to Carbapenem antibiotics in page 2, line 46?

Response: The mechanism of mutation is added in lines 47-56.

Question 2) Author used Gold nanoparticle in this work. How is it cost effective compared to other metal nanoparticle?

Response: Other nanoparticles are available but are unstable and toxic. GNPs is benign to humans and environment and are stable for >3 years, making them cost-effective. This information was added in lines 151-153.

Question 3) Can author mentioned how long the GNPs aggregation time is prevent in presence of target gene and maintain it’s colour?

Response: The GNPs are prevented from aggregation for at least 30 min. This information is added in line 341.

Question 4) Based on the method described in the manuscript, this diagnostic tool require many pre-preparation steps such whole genome extraction, chemicals preparation to stabilize GNP. How it will be cost effective than other existing methods?

Response: Our method has only 3 steps after extraction of DNA; adding DNA to GNPs/Probe, placing in thermocycler, and HCl application. This information was mentioned in lines 523-525. Initial cost estimate for our method is < $2 as explained in line 526. The cost-effectiveness of our biosensor is further explained in lines 527-530.